# Heparan Sulfate Is a Cellular Receptor for Enteric Human Adenoviruses

**DOI:** 10.3390/v13020298

**Published:** 2021-02-14

**Authors:** Anandi Rajan, Elin Palm, Fredrik Trulsson, Sarah Mundigl, Miriam Becker, B. David Persson, Lars Frängsmyr, Annasara Lenman

**Affiliations:** 1Section of Virology, Department of Clinical Microbiology, Umeå University, 90185 Umeå, Sweden; elin.palm@umu.se (E.P.); c.f.b.trulsson@lumc.nl (F.T.); sarah.mundigl@outlook.de (S.M.); miriam.becker@umu.se (M.B.); david.persson@sva.se (B.D.P.); lars.frangsmyr@umu.se (L.F.); anna-sara.lenman@umu.se (A.L.); 2The Laboratory for Molecular Infection Medicine Sweden (MIMS), Umeå University, 90185 Umeå, Sweden; 3Department of Medical Biochemistry and Cell Biology, University of Gothenburg, 41390 Gothenburg, Sweden; 4Department of Cell and Chemical Biology, Leiden University Medical Center, 2333 ZC Leiden, The Netherlands; 5Institute for Experimental Virology, TWINCORE, Centre for Experimental and Clinical Infection Research, 30625 Hannover, Germany; 6National Veterinary Institute, SVA, 75189 Uppsala, Sweden

**Keywords:** enteric adenovirus, heparan sulfate, short fibers, capsid proteins, fiber knobs

## Abstract

Human adenovirus (HAdV)-F40 and -F41 are leading causes of diarrhea and diarrhea-associated mortality in children under the age of five, but the mechanisms by which they infect host cells are poorly understood. HAdVs initiate infection through interactions between the knob domain of the fiber capsid protein and host cell receptors. Unlike most other HAdVs, HAdV-F40 and -F41 possess two different fiber proteins—a long fiber and a short fiber. Whereas the long fiber binds to the Coxsackievirus and adenovirus receptor (CAR), no binding partners have been identified for the short fiber. In this study, we identified heparan sulfate (HS) as an interaction partner for the short fiber of enteric HAdVs. We demonstrate that exposure to acidic pH, which mimics the environment of the stomach, inactivates the interaction of enteric adenovirus with CAR. However, the short fiber:HS interaction is resistant to and even enhanced by acidic pH, which allows attachment to host cells. Our results suggest a switch in receptor usage of enteric HAdVs after exposure to acidic pH and add to the understanding of the function of the short fibers. These results may also be useful for antiviral drug development and the utilization of enteric HAdVs for clinical applications such as vaccine development.

## 1. Introduction

The *Adenoviridae* family contains more than 100 types of human adenoviruses (HAdVs), which are classified into seven species (A to G) based on serum neutralization and genome sequence analyses [1,2]. HAdVs cause different types of infections such as ocular (species B, D and E), respiratory (species A, B, C and E) and enteric infections (species F) [3]. HAdV-F40 and -F41 are the only members of species F and are referred to as enteric human adenoviruses. They cause gastrointestinal infections primarily in children below five years of age [4,5]. With developments in molecular diagnostics, enteric HAdV-F40 and -F41 are now recognized as a leading cause of gastroenteritis and diarrhea-associated mortality in young children [6,7]. The HAdV particle consists of three major capsid proteins: fiber, penton base and hexon. The protruding fiber is needed for initial attachment to host cell receptors [8,9], followed by secondary binding of the penton base to integrins leading to internalization of the virus into the cells [10,11,12]. Several attachment receptors have been identified for HAdVs, including the Coxsackievirus and adenovirus receptor (CAR) [8], CD46 [13,14,15], desmoglein 2 [16] and sialic acid-containing glycans [9,17,18]. HAdV-C2 and -C5 [19,20] and, to some extent, HAdV-B3 and -B35 [21] bind heparan sulfate as receptors/coreceptors to facilitate infection. Strikingly, sulfated glycosaminoglycans (GAGs) can also act as decoy receptors for sialic acid-binding HAdV-D37, where binding of the virus to GAGs prevent efficient infection [22]. Thus, GAGs play an important role for several HAdV types, but have not been associated with enteric adenoviruses until now. HS and other GAGs such as chondroitin sulfate (CS), dermatan sulfate (DS), keratan sulfate (KS) and hyaluronic acid (HA) are long, unbranched polysaccharides consisting of repeating disaccharide units of an amino sugar (N-acetylglucosamine or N-acetylgalactosamine) and an uronic sugar (iduronic or glucuronic acid) or galactose [23].

Enteric HAdV-F40 and -F41 have two different types of fibers, long fibers (LF) and short fibers (SF), a characteristic shared only with HAdV-G52 and no other human adenovirus [9,24,25]. HAdV-G52 was isolated from a small outbreak of gastroenteritis [26], and, although morphologically similar to HAdV-F40 and -F41, this virus has been classified into a new species—species G. In HAdV-F40 and -F41, the SF is more abundant in the capsid than the LF with a ratio of 6:1 [25]. The LF binds CAR [8], but no cellular interaction partners have been identified for the SFs. The SF of HAdV-F41 stimulates enteroendocrine cells of the small intestine, called enterochromaffin cells, to secrete serotonin, which activates enteric glial cells causing diarrhea and vomiting [27].

Unlike most other HAdVs, which can infect cells of multiple organs, enteric HAdVs exclusively infect the intestinal tract and do not cause any infections at other sites. We recently showed that this restricted tropism can be attributed to differences in the external structure of the virion as compared to respiratory and ocular HAdVs [28]. The restricted tropism has also been suggested to depend on the ability of the SF of HAdV-F40 and HAdV-F41 to protect the virus against acidic gastric conditions [29,30]. Here, we identify HS as a cellular attachment factor for the SF. Our findings also suggest a switch in receptor usage after exposure to acidic pH from being LF:CAR dependent to SF:HS dependent, further supporting the theory that SFs confer the enteric tropism of these viruses.

## 2. Materials and Methods

### 2.1. Cells, Viruses and Antibodies

Cells: A549 cells (a kind gift from Dr. Alistair Kidd) were cultured in Dulbecco’s Modified Eagle Medium (DMEM; Sigma-Aldrich, St. Louis, MO, USA) with 5% fetal bovine serum (FBS; Thermo Fisher Scientific, Waltham, MA, USA), 20 mM HEPES (Sigma-Aldrich, St. Louis, MO, USA) and 20 U/mL penicillin +20 µg/mL streptomycin (PEST; Thermo Fisher Scientific, Waltham, MA, USA); HAP1-WT, HAP1-ΔCAR (Horizon Discovery, Waterbeach, UK) and HAP1-ΔGAG (beta-1,3-galactosyltransferase deficient; a kind gift from Dr. Frank Kuppeveld) were cultured in Iscove’s Modified Dulbecco’s Medium (IMDM; Gibco, Thermo Fischer Scientific, Waltham, MA, USA) with 10% FBS and PEST; CHO-K1, and CHO-ΔHS (pgsD-677) [31] were cultured in Ham’s F12 (Sigma-Aldrich, St. Louis, MO, USA) with 10% FBS and PEST. HuTu80 (ATCC, Manassas, VA, USA) cells were grown in DMEM with PEST and 10% FBS; Pro-5 (LGC Promochem, Teddington, UK) and Lec2 cells (LGC Promochem, Teddington, UK) were grown as described previously [32,33]. Human neuroblastoma SK-N-SH cells (LGC Promochem, Teddington, UK) were grown in DMEM supplemented with 10% FBS, 20 mM HEPES and PEST. SH-SY5Y cells (LGC Promochem, Teddington, UK) were grown in DMEM:Ham’s-F12 (Sigma-Aldrich, St. Louis, MO, USA ) at 1:1, with the same supplements as the parental SK-N-SH cell line.

Viruses/Vector: HAdV-F40 (strain Hovix), HAdV-F41 (strain Tak), HAdV-D36 (strain 275), HAdV-D37 (strain 1477), HAdV-C5 (strain adenoid 75) were grown in A549 cells with or without ^35^S-labeling as described previously in [34] except that the elution buffer after NAP column (GE Healthcare, Chicago, IL, USA) purification and storage buffer was phosphate buffered saline (PBS). Virions were stored in PBS with 10% glycerol at −80 °C. HAdV-F41 GFP vector was produced as described in [35].

Antibodies: anti-RGS His mouse monoclonal antibody (Qiagen, Hilden, Germany) and Alexa Fluor 488 donkey antimouse IgG (Life technologies, Carlsbad, CA, USA) were used in flow cytometry to detect recombinant fiber knobs. Serotype-specific rabbit polyclonal antisera against HAdV-C5, -D36, -D37, -F40 and -F41 were a kind gift from Dr. Göran Wadell [36]. Anti-HS (clone F58-10E4, Amsbio, Abingdon, UK) and anti-CAR (clone RmcB, Sigma-Aldrich, St. Louis, MO, USA) monoclonal antibodies were used to check expression levels of HS and CAR, respectively.

### 2.2. Recombinant Fiber Knobs, Enzymes, Metabolic Inhibitors, Soluble GAGs and Cholera Toxin

Recombinant FKs: Cloning, expression and purification of 40SFK (amino acids 215 to 387), 40LFK (amino acids 348 to 547), 41LFK (amino acids 363 to 562), 5FK (amino acids 387 to 581) and 37FK (amino acids 172 to 365) were performed in a similar manner as described in [9] and 52SFK was produced as described in [9].

Enzymes: *Vibrio cholera* neuraminidase (Sigma-Aldrich, St. Louis, MO, USA), fig tree latex ficin (Sigma-Aldrich, St. Louis, MO, USA ), *Engyodontium album* proteinase K (Sigma-Aldrich, St. Louis, MO, USA) proteases and *Flavobacterium heparinum* heparinase III (Sigma-Aldrich, St. Louis, MO, USA).

Metabolic inhibitors: P4 [(1R,2R)-1-phenyl-2-hexadecanoylamino-3-pyrrolidino-1-propanol] and inactive enantiomer of P4 (1S,2S, both kindly provided by Dr. Ronald L. Schnaar).

Soluble GAGs: Heparin (from porcine intestinal mucosa; Sigma-Aldrich, St. Louis, MO, USA), chondroitin sulfate A (from bovine trachea; Sigma-Aldrich, St. Louis, MO, USA), dermatan sulfate (from porcine intestinal mucosa; Sigma-Aldrich, St. Louis, MO, USA), chondroitin sulfate mix A and C (from shark cartilage; Sigma-Aldrich, St. Louis, MO, USA) and hyaluronic acid (from *Streptococcus equi*; Sigma-Aldrich, St. Louis, MO, USA). Alexa Fluor 488 conjugated cholera toxin subunit B, which binds the ganglioside GM1 (Molecular probes, Invitrogen, Carlsbad, CA, USA).

Recombinant human CAR (CXADR Fc chimera full length extracellular D1D2 domain; R&D systems, Minneapolis, MN, USA).

### 2.3. Fiber Knob Binding Assay

FK binding to cells was evaluated by flow cytometry. Cells (A549, CHO-K1, CHO-ΔHS, Pro-5, Lec2, SK-N-SH, SH-SY5Y or HuTu80) were detached from culture flasks by PBS-EDTA (0.05% EDTA), counted and then reactivated in growth media for one hour at 37 °C under shaking conditions. After this incubation, the cells (2 × 10^5^ cells/mL) were added to a V-bottom 96-well plate and washed with serum-free media. We added 10 µg/mL of fiber knobs in serum-free media to the cells and incubated for one hour at 4 °C. The cell and fiber knob mix was washed with FACS buffer (PBS with 2% FBS) to remove unbound fiber knobs. To detect bound fiber knobs, cells were first incubated with anti-RGS His antibody (diluted 1:200 in FACS buffer) for 30 min at 4 °C, washed and then incubated with Alexa Fluor 488 donkey antimouse IgG antibody for another 30 min at 4 °C. After a final wash, samples were analyzed using FACSLSRII flow cytometer (Becton Dickinson, Franklin Lakes, NJ, USA) and results were analyzed by FACSDiva software (Becton Dickinson, Franklin Lakes, NJ, USA). The results were expressed as geometrical mean of Alexa Fluor 488 fluorescence. FK binding experiment was performed with the following variations: Before addition of the FKs, cells were incubated with or without (a) different concentrations of ficin and proteinase K for 30 min at 37 °C to degrade proteins on the cell surface (b) 2.5 µM each of P4 or inactive P4 for 5 days at 37 °C to inhibit glycolipid biosynthesis via the glycosylceramide synthase enzyme as described in [37], (c) 10 mU/mL of *Vibrio cholera* neuraminidase for one hour at 37 °C to remove cell surface sialic acids, or (d) 1 U/mL of *Flavobacterium heparinum* heparinase III for one hour at 37 °C to degrade HS. Another variation to this experiment was (e) the preincubation of fiber knobs with a wide range of concentrations of soluble GAGs for one hour at 4 °C before addition to cells.

### 2.4. Infection Assay

A549, CHO-WT, CHO-ΔHS or HuTu80 cells were grown as monolayers overnight at 37 °C in 96-well clear bottom plates (Greiner bio-one, Kremsmünster, Austria). For experiments blocking infection with soluble GAGs, the cell monolayers were washed three times with serum-free medium and incubated with heparin-treated virions for one hour on ice. Unbound virions were removed by washing the cells three times with serum-free medium. Medium with 1% FBS was then added and the cells were incubated for 44 h at 37 °C. Next, the cells were washed with PBS and fixed with ice-cold methanol for 10 min at −20 °C. Viral capsid proteins were stained with polyclonal rabbit antibodies diluted in PBS for 30 min at room temperature. The cells were washed twice for 15 min with PBS and then incubated with Alexa Fluor 647 goat antirabbit IgG (H+L) secondary antibody for detection of infected cells, and Hoechst 33342 (Thermo Fisher Scientific, Waltham, MA, USA) for visualization of cellular nuclei. After two more washes with PBS, number of transfected cells were analyzed using the Tina program of the TROPHOS (Luminy Biotech Enterprises, Marseille, France). For transduction with HAdV-F41 vector, the vector was incubated with CHO cells for two hours on ice before adding medium with 1% FBS and incubating the cells at 37 °C for 72 h. The cells were given a final PBS wash before analyzing them on the TROPHOS.

### 2.5. ^35^S-Labeled Virion Binding Assay

Cells were detached with PBS-EDTA, reactivated in growth medium for one hour at 37 °C, pelleted in V-bottom 96-well plates (2 × 10^5^ cells/mL) and then washed with serum-free media. We preincubated 2 × 10^9 35^S-labeled virions with or without different concentration of heparin in serum-free media for one hour at 4 °C and then added to the cells in the plate and incubated for another hour at 4 °C (all steps were done in suspension). Unbound virions were washed away with PBS and cell associated radioactivity was measured in a Wallac 1450 Microbeta liquid scintillation counter (Perkin Elmer, Waltham, MA, USA). In the experiment with HAP1 cells, cells were first incubated with 2 µg/mL of 41LFKs for one hour on ice. Meanwhile, the virions were treated with simulated gastric fluid without enzyme (pH 1.1–1.3; 2.0 g/l NaCl and 3.0 g/l HCl; based on synthetic gastric fluid formulation by Ricca Chemical Company catalog #7108-16) or incubated with plain DMEM for five minutes at 37 °C, and then neutralized with 20× volume of serum-free media. Virions were subsequently incubated with or without 100 µM soluble heparin for one hour on ice. The fiber knobs were washed away from the cells with serum-free media and the virus-heparin mixture was added to the cells for one hour on ice.

### 2.6. HAdV Uptake Assay

Virus labeling: Purified HAdV-F40, -F41 and -C5 virions were labeled on free amine groups with Alexa Fluor 488 NHS ester (Thermo Fisher Scientific, Waltham, MA, USA) by incubating with a 10-fold molar excess of the dye in PBS, pH 7.4, for one hour under rotation. Virions were then separated from free dye by CsCl gradient centrifugation as in the initial purification described under Viruses/Vectors, with subsequent desalting on NAP columns and elution in PBS, pH 7.4. Virions were supplemented with 4% glycerol and frozen at −80 °C.

Imaging-based HAdV entry assay: A549 cells were seeded in 8-well µ-slides (Ibidi, Gräfelfing, Germany; 45,000 cells/well) 24 h prior to experimentation. Alexa Fluor 488-labeled HAdVs were preincubated with 0 µM, 100 µM or 1000 µM soluble heparin in medium without FBS on ice for one hour. The virus-heparin mixtures were then added to the cells and incubated for one hour on ice. Unbound virions were removed, and cells were supplemented with growth medium to allow for virus entry for five hours at 37 °C. The cells were then transferred back on ice, washed with cold PBS, blocked with cold PBS containing 3% BSA for 15 min and stained for remaining extracellular virus particles using anti-Alexa Fluor 488 antibody (Thermo Fisher Scientific, Waltham, MA, USA) and antirabbit Alexa Fluor 568 antibody (Thermo Fisher Scientific, Waltham, MA, USA) in PBS with 3% BSA for 45 min each. Cells were washed with cold PBS and subsequently fixed with 4% PFA at room temperature for 20 min. Cells were then stained with Hoechst 33342 (Thermo Fisher Scientific, Waltham, MA, USA) and Alexa Fluor 647-labeled wheat germ agglutinin (WGA; Thermo Fisher Scientific, Waltham, MA, USA) for visualization of cellular nuclei and plasma membrane, respectively. Images were acquired on a Leica SP8 confocal microscope with a 63× oil objective at the BICU imaging unit of Umeå University. Sufficient fields of view were acquired for each condition in independent experiments to reach a total of more than 100 cells. For HAdV-C5 and HAdV-F40, single medial focal planes from random fields of view were acquired. For HAdV-F41, cell spanning stacks of 10 slices with 0.5 µm step size from random fields of view were imaged and converted by maximum intensity projection (Fiji; [38]). Internalization ratio was determined by primary object identification in CellProfiler (version 3.1.9; [39]) as follows: cell outlines were determined using the cell counter stain WGA. Cell mask was transferred onto the channels for all virus (Alexa Fluor 488) and extracellular virus (Alexa Fluor 568). Within the masked images, virus particles per cell area were identified by primary object detection.

### 2.7. Surface Plasmon Resonance

SPR experiments were performed using BIAcore T200 optical biosensors (GE Healthcare, Chicago, IL, USA). CAR was covalently immobilized to CM5 sensor chips by standard EDC/NHS coupling for 420 s at a flow rate of 10 μL/minute using a 50 μg/mL concentration in 10 mM sodium acetate, pH 4.0. For heparin interaction a heparin chip (Xantec, Düsseldorf, Germany) was used. For each BIAcore kinetic experiment, HAdV-F40, -F41 or 40SFK was pretreated with or without acidic pH as described under ^35^S-labeled virion binding assay, followed by a buffer exchange to running buffer (20 mM sodium phosphate, 150 mM NaCl, with 0.05% (*v*/*v*) surfactant P20, pH 7.4) and injected for 120 s at 30 μL/minute followed by 120 s of dissociation. All covalent surfaces were regenerated with one 30 s pulse of 10 mM glycine–HCl (pH 1.5, GE Healthcare, Chicago, IL, USA). All experiments were conducted at 25 °C. Virus binding to CAR was corrected for binding to an empty channel. Sensorgrams were calculated and processed using BIAcore T200 evaluation software (version 2.0, GE Healthcare, Chicago, IL, USA).

### 2.8. Statistical Analysis

The results are expressed as mean ± standard error of mean (SEM) and either t-test or one-way ANOVA with Dunnett’s multiple comparison’s test was performed using GraphPad Prism version 8.2.1. *p*-values < 0.05 were considered statistically significant. All experiments were performed at least three times with duplicate samples.

## 3. Results

### 3.1. HAdV-F40 Short Fiber Knob Binding to Cells Requires Heparan Sulfate-Containing Proteins

The knob domain of the fiber protein (FK) mediates the attachment of HAdVs to their cellular receptors on host cells. Assuming that the short fiber knob (SFK) of enteric HAdVs also contributes to cell attachment, we first characterized the nature of HAdV-F40 SFK interaction partners by flow cytometry using A549 cells, a cell line that supports productive infection of enteric HAdVs [40]. To test for dependence of proteins and glycolipids, we first treated cells with either ficin, a cysteine endopeptidase or proteinase K, a broad-spectrum serine protease to remove cell surface proteins, or with P4, an inhibitor of *de novo* glycolipid biosynthesis. Treatment with ficin and proteinase K reduced binding of 40SFK by more than 80% (Figure 1A), while the glycolipid biosynthesis inhibitor P4 had no effect on binding of any of the FKs used (Figure 1B). We used CAR-binding HAdV-C5 fiber knob (5FK) and HAdV-F40 long fiber knob (40LFK) as controls for the protease treatment and the glycolipid-binding cholera toxin subunit B as a control for P4 treatment. As expected, protease treatment reduced 5FK and 40LFK binding and P4 treatment diminished binding of cholera toxin. Both 5FK and 40LFK bind CAR, but we observed a remarkable difference in sensitivity to protease treatment between the two, with 40LFK being substantially more sensitive than 5FK. In addition, we observed an increase in 5FK binding at low proteinase K concentrations that could potentially result from the partial removal of the protein factors from the cell surface revealing novel binding sites for 5FK. We did not evaluate these results further as it did not fall within the scope of this study. The strong reduction observed for 40SFK binding after protease treatment indicated that a protein component is required for its efficient binding to A549 cells. Since P4 did not inhibit binding of 5FK, 40SFK or 40LFK, we concluded that glycolipids are not a target for these FKs. Next, we investigated whether the 40SFK:cell interaction was a pure protein-protein interaction or if glycans could also be involved. Along with being receptors for other HAdV types, sialic acid-containing glycans [41] and HS proteoglycans [20] are abundantly expressed in the intestinal glycocalyx [42,43], which makes them ideal attachment factors for viruses attacking from the lumen. We treated A549 cells with *V. cholerae* neuraminidase, which cleaves all sialic acids, and evaluated the binding of FKs. Neuraminidase pretreatment reduced the binding of 40SFK by 30% and 40LFK by 10% (Figure 1C), while the sialic acid-binding HAdV-G52 short fiber knob (52SFK) control [9] was reduced by more than 85%. Since neuraminidase reduced both 40SFK and 40LFK binding to A549 cells, we also analyzed FK binding to sialic acid-expressing Pro-5 cells and Pro-5-derived, sialic acid-lacking Lec2 cells [32,33]. We observed no significant reduction in 40SFK binding to neuraminidase-treated Pro-5 cells as compared to untreated Pro-5 cells. However, a slightly reduced binding was detected to sialic acid-deficient Lec2 cells as compared to sialic acid-expressing Pro-5 cells (Figure 1D). As expected, a more significant reduction was seen for the sialic acid-binding 52SFK control, while the CAR-binding 5FK and 40LFK displayed no or low binding, respectively. Since 52SFK, the only other SFK found in HAdVs, preferentially binds α2,8-linked polysialic acid [18], we also evaluated the binding of 40SFK to polysialic acid-expressing SHSY-5Y cells and the polysialic acid-deficient parental SK-N-SH cell line. We found that 40SFK bound to both cell types to a similar extent (Figure 1E). Taken together, these results suggested that sialic acid-containing glycans are not important attachment factors for 40SFK.Next, we treated A549 cells with heparinase III from *Flavobacterium heparinum*, which cleaves cellular HS. The removal of HS drastically reduced 40SFK binding, whereas there was little effect on the other FKs (Figure 1F). An anti-HS antibody bound substantially less efficiently to heparinase III-treated cells as compared to nontreated cells, demonstrating that heparinase III treatment was efficient in cleaving HS from the cell surface. The reduced binding after both protease and heparinase III treatment of target cells suggested that HS-containing proteins could function as attachment factors for 40SFK on the cell surface.

### 3.2. HAdV-F40 SFK Binding to A549 Cells Requires Sulfated GAGs

To validate the effect of heparinase III treatment, and to determine the potential specificity of 40SFK for specific GAG types, we next investigated binding of 40SFK to A549 cells in the presence of different, soluble GAGs potentially blocking cell binding. Flow cytometry analysis of 40SFK binding after preincubation with soluble GAGs demonstrated that all GAGs except HA reduced 40SFK binding but with varying efficiency (Figure 2A). Heparin (a soluble analogue of HS) most efficiently inhibited binding with an inhibitory concentration (IC50) as low as 1 nM. CS and DS also inhibited 40SFK binding but were less efficient, being 50 and 100 times higher IC50, respectively, as compared to heparin. As expected, we did not observe any significant inhibition of GAGs on 5FK binding to cells (data not shown) since the domain of HAdV-5 that interacts with GAG is suggested to be present on the fiber shaft and not the knob domain [20]. The inability of HA—a GAG structurally similar to HS lacking sulfate groups—to decrease 40SFK binding indicated that sulfation of GAGs was needed for efficient interaction. To further validate the interaction with HS, we studied knob binding to CHO cells lacking or expressing HS. We observed that 40SFK binding to CHO-ΔHS cells (expressing all GAGs except HS) was completely depleted as compared to the parental control cell line CHO-K1 (Figure 2B). Neither 5FK nor 40LFK bound efficiently to these cells, which was expected since these cells do not express CAR [44]. An anti-HS antibody was used to verify the absence of HS on CHO-ΔHS. In summary, these findings show that 40SFK interact with sulfated GAGs, preferably with HS.

### 3.3. Sulfated GAGs Play an Important Role in HAdV-F40 and -F41 Infection of A549 Cells

Since soluble, sulfated GAGs reduced 40SFK binding to A549 cells, we tested their effect on HAdV infection of A549 cells. Due to the close relationship between HAdV-F40 and -F41 SFKs, we hypothesized that GAGs may be of importance for both viruses and therefore included HAdV-F41 in the study. To investigate the importance of HS for these viruses, we first analyzed the effect of preincubating wild type (WT) viruses with different concentrations of heparin in infection experiments. A dose-dependent decrease in HAdV-F40 and -F41 infection was observed, with about 80% reduction at the highest concentration (Figure 3A). Heparin did not affect HAdV-C5 or HAdV-D36 infection but inhibited HAdV-D37, -F40 and -F41 infection with comparable efficiency. HS-binding HAdV-D37 was included as a positive control [22,36]. The reduction of HAdV-F40 and -F41 infection after preincubation with heparin indicated that the enteric HAdVs required an interaction with HS for infection of A549 cells. To further examine the SFK:HS interaction without interference from the LFK:CAR interaction, we used CHO-cells since they do not express human CAR. We transduced CHO-K1 and CHO-ΔHS cells with a GFP-expressing HAdV-F41 vector since CHO cells do not support viral hexon production used to quantify infection by WT viruses. Here, we observed an 80% decrease in transduction of the cells lacking HS (Figure 3B). No difference in transduction levels of CHO-ΔHS compared to CHO-K1 was observed with a GFP-expressing HAdV-C5 vector (data not shown). Additionally, upon pretreatment of HAdV-F41 vector with heparin, transduction was lowered to the same amount as in the CHO-ΔHS cells (Figure 3C). Taken together these results show that HS is important for the infection of cells by both HAdV-F40 and -F41.

### 3.4. HAdV-F40 and -F41 Binding to and Uptake in A549 Cells Is Not Affected by Heparin

As cell surface HS serves as an attachment factor for the SFKs of enteric HAdVs, we next analyzed the role of HS during virion binding to A549 cells by preincubating ^35^S-labeled HAdV-F40 or -F41 virions with soluble heparin before attachment to cells. Surprisingly, we did not see a decrease in HAdV-F40 or -F41 virion binding in the presence of heparin, not even at a concentration of 1 mM (Figure 4A). This was peculiar since 40SFK binding to A549 cells was completely blocked with as little as 10 nM of heparin (Figure 2A), and infection was reduced with 100 µM heparin (Figure 3A). As expected, the binding of the control virions to A549 cells was reduced in a dose-dependent manner (HAdV-D37) or was unaffected (HAdV-C5) (Figure 4A).

To investigate if HS was involved in virion internalization, Alexa Flour 488-labeled HAdV-F40 and -F41 virions were preincubated with heparin followed by a synchronized uptake into A549 cells for five hours at 37 °C, after which the number of extracellular and intracellular virions were quantified by confocal microscopy. After five hours at 37 °C, approximately 50% of HAdV-F40 and 20% of HAdV-F41 virions were internalized into the cells (Figure 4B). Preincubation with heparin did not reduce internalization even at high concentrations. Almost all HAdV-C5 virions were internalized regardless of heparin treatment (Figure 4B). Since heparin did not affect the number of bound or internalized HAdV-F40 and -F41 virions, these data suggest that the SFK:HS interactions do not contribute to virion attachment or internalization at neutral pH.

### 3.5. LFK:CAR-Dependent Cell Attachment of Enteric HAdVs Is Switched to SFK:HS-Dependent Cell Attachment after Virion Exposure to Acidic pH

Previous studies on SFs of enteric HAdVs suggest that the SF is important for the enteric tropism of these viruses since it imparts resistance to acidic pH [29,30]. To further understand the role of the SFK:HS interaction during enteric HAdV infection, we addressed the potential impact of acidic pH on virion attachment to host cells. We analyzed the binding of ^35^S-labeled virions to WT haploid HAP1 cells, derived from a human chronic myelogenous leukemia cell line and to HAP1 cells devoid of CAR or GAGs. The absence of CAR and HS from these cells was verified by flow cytometry using anti-CAR and anti-HS monoclonal antibodies (data not shown). At neutral pH (Figure 5A), HAdV-F40 bound with similar efficiency to HAP1 WT cells and HAP1-∆GAG cells, but to a substantially lower degree to HAP1-∆CAR cells, suggesting that at this pH, HAdV-F40 virions bound to HAP1 cells mainly in a LF:CAR dependent manner, and did not rely on SF:GAG interactions. This was further supported by the observation that the preincubation of cells with soluble, CAR-binding 41LFK reduced binding of HAdV-F40 virions to CAR-expressing cells, while the preincubation of virions with heparin did not affect binding to these cells (Figure 5A). After the exposure of virions to acidic pH (pH 1.1 to 1.3) (Figure 5B) the binding of HAdV-F40 to WT HAP1 cells remained high, while binding to HAP1-∆GAG cells decreased and binding to HAP1-∆CAR cells increased, as compared to binding at neutral pH. Interestingly, at acidic pH the preincubation of virions with heparin reduced HAdV-F40 binding to both HAP1 WT cells and HAP1-∆CAR cells efficiently (Figure 5B).

The binding to HAP1-∆GAG cells in the presence of heparin was also reduced, but to a lesser extent. On the other hand, the preincubation of cells with 41LFK did not affect acidic pH-exposed virion binding. Similar results were observed when these experiments were performed with HAdV-F41 (Figure 5C,D). Taken together, these results suggested that virion exposure to acidic pH changes the mechanism by which enteric HAdVs attach to host cells. The LFK:CAR interaction dominates in a neutral environment, but exposure to acidic pH switches the mechanism of virion binding to be SFK:HS dependent.

### 3.6. Acidic pH-Exposed HAdV-F40 and -F41 Virions Lose CAR-Binding Capacity But Retain HS-Binding Capacity

To validate the HAdV-F40 and -F41 preference for HS binding after exposure to acidic pH, we performed surface plasmon resonance (SPR) studies with CAR or heparin attached on the sensor chip and virions as the analyte. At neutral pH, both HAdV-F40 and -F41 virions displayed a very strong binding to CAR, which was almost completely abrogated after exposure to acidic pH (Figure 6A–D). On the other hand, virion binding to heparin was maintained at similar levels before and after acidic pH treatment (Figure 6E–H). When the binding between 40SFK and heparin was tested (Figure 6I), we observed a substantially increased binding after exposure of 40SFK to acidic pH (Figure 6J). These results support our theory that acidic pH exposed HAdV-F40 and -F41 lose their ability to bind CAR, but instead bind HS on target cells through an interaction with the SF.

### 3.7. Heparin Specifically Reduces Infection and Short Fiber Knob Binding on a Small Intestinal Cell Line

HAdV-F40 and -F41 cause infection in the small intestine [45]. To validate the role of HS in a more physiologically relevant cell line, we studied the importance of the SFK:HS interaction using a small intestinal cell line, HuTu80. The results obtained with HuTu80 are largely consistent with those obtained using A549 cells. Preincubation with soluble heparin reduced 40SFK binding to HuTu80 cells very efficiently with more than 50% of the binding reduced at 1 nM heparin (Figure 7A). As expected, heparin also inhibited 37FK binding but not that of 40LFK or 5FK. We also observed that 40SFK binding was inhibited by CS, but only at a much higher concentrations (Figure 7B), and not at all by HA (Figure 7C) suggesting that 40SFK interacts mainly with HS on these cells. Finally, the preincubation of virions with heparin also reduced HAdV-F40 and -F41 (and positive control HAdV-D37) infection of HuTu80 cells but did not influence infection of the negative control HAdV-C5 (Figure 7D). Neither CS (Figure 7E) nor HA (Figure 7F) reduced HAdV-F40 and -F41 infection as efficiently as heparin. Taken together, these results demonstrate that enteric HAdVs interact with HS through their short fibers on multiple cell lines, including small intestinal cells.

## 4. Discussion

A distinguishing feature of enteric HAdVs is the presence of SFs in the viral capsid. The relatively high ratio of SFs to LFs suggests that SFs play an important role in the infectious cycle of these viruses. Despite being a leading cause of diarrhea and diarrhea-associated mortality in young children, no cellular attachment factors have been identified for the SFs of enteric HAdVs. Here, we demonstrate for the first time that HS is an important host factor for infection by enteric HAdVs. We pinpoint this interaction to occur through the SF by the observations that SF does not bind—(i) when cells are treated with heparinase III, (ii) in the presence of heparin and (iii) when cells are devoid of HS. We further demonstrate that enteric HAdVs undergo a switch in receptor usage when exposed to pH mimicking the acidic environment in the stomach. Exposure of enteric HAdVs to acidic pH resulted in a loss of LFK-dependent CAR-binding capacity but maintained SFK-dependent HS-binding to cells. Since enteric HAdVs are transmitted through the fecal–oral route by the ingestion of contaminated food or water, we propose a novel function of the short fiber, where it compensates for the loss of CAR-binding occurring during passage through the harsh acidic environment of the digestive tract and facilitates a first round of infection when reaching the permissive, HS-expressing cells in the small intestine. Subsequent rounds of infection result in production and release of progeny virions to an environment with neutral pH, where the LFK:CAR interaction can be of importance for further spread of the infection. It has previously been shown that cells infected with CAR-binding and CD46 binding HAdVs produce an excess of soluble fibers, which upon release can disrupt intercellular CAR and CD46 homodimers and destroy the integrity of tight junctions, thereby facilitating spread of progeny virions [46,47]. It is possible that the LF of HAdV-F40 and -F41 could exert a similar function and enable the transmission of the released progeny virions within a tissue, or between tissues. The released virions will not be subjected to acidic pH and will therefore maintain their ability to bind CAR in order to further propagate the infection. Exposure to gastrointestinal fluids is very likely to cause structural alterations in virus particles, making them noninfectious or, as for rotavirus, activating them to enable infection. Rotavirus requires proteolytic cleavage of proteins in the outer capsid layer by enzymes present in digestive fluids to render it more infectious [48]. For enteric HAdVs, the harsh environment of the stomach may destroy the LF:CAR interaction that would make the virions less infectious without the rescuing effect of the SF:HS interaction. Further studies are needed to address the exact molecular mechanism behind the effect of acidic gastric juice on SFs.

The ability of 40SFK to bind cellular HS was efficiently blocked with nanomolar concentrations of heparin. A significantly higher concentration was needed to reduce, but not completely block, HAdV-F40 and -F41 infection. This can partly be explained by the presence of the LF in the virion and its ability to interact with CAR as a receptor for attachment and entry, independent of the SF. To our surprise neither virion binding nor entry was affected by preincubation with heparin at neutral pH, indicating that the SF:HS interaction is not of importance for these steps in the viral life cycle under physiological conditions. Nevertheless, we observed a specific reduction of infection by heparin that could not be seen with CS or HA: nor could it be explained by a general negative effect on the cells since no effect was observed on the control viruses HAdV-C5 and HAdV-D36. We speculate that under neutral pH, the intact LF:CAR interaction is mainly responsible for virus entry and that heparin bound to the SF may block a subsequent interaction of the SF with an intracellular protein needed for efficient infection. This theory is supported by the suggestion that 41SF can interact with a novel intracellular protein, ParAd41, which may facilitate attachment to the nuclear membrane prior to injection of DNA to the nucleus [49].

The only other HAdV to have a SF is HAdV-G52 [26]. The SF of HAdV-52 binds to long negatively charged glycan chains of polymerized sialic acid residues known as polysialic acid [18]. We show here that 40SFK does not bind polysialic acid as indicated by similar binding levels to cells expressing and lacking polysialic acid. Instead 40SFK cellular binding shows a dependency of sulfated GAGs, preferably HS. Polysialic acid is a posttranslational modification present on proteins such as NCAM [50], SynCAM-1 [51], neuropilin-2 [52] and CCR7 [53]. Since these proteins are absent or only expressed in low amounts on target cells of the small intestine, they are unlikely receptor candidates for the SF of enteric HAdVs. Although HAdV-G52 was isolated from a case of gastroenteritis, it was classified in a separate species and is not considered one of the enteric HAdVs [26], which is the opposite of HAdV-F40 and -F41, which are common in the human population [54,55]. HAdV-G52 is rarely isolated [56] and even though it is closely related to HAdV-F40 and -F41 it is actually more similar to simian AdVs [26]. An interesting observation on the divergence of species F and G, both containing short fibers, from the other HAdVs species, is the evolved ability of short fiber binding to negatively charged glycans. While 52SFK has evolved to bind polysialic acid, 40SFK and 41SFK binds heparan sulfate. It still needs to be explored whether exposure to acidic pH causes a receptor switch for HAdV-G52 as well.

The precise function of the short fibers of enteric HAdVs has been under investigation for a long time. The 41SFK is suggested to function as a viral enterotoxin by stimulating release of serotonin from enterochromaffin cells which could result in symptoms such as vomiting and diarrhea. Furthermore, the SF of enteric HAdVs have been suggested to be responsible for the restricted intestinal tropism since these viruses were shown to be more resistant to acidic pH than other HAdV types [29,30]. This claim was refuted by another study describing that a recombinant HAdV-C5 vector carrying the 41SFs showed reduced affinity for enterocytes [57]. The latter study, however, did not subject the virions to acidic pH, which could have affected their results and interpretations. Here, we identified HS as an interaction partner to the SF and explained its role in cellular attachment after exposure to acidic pH. In our opinion, the SF contributes to infection and pathogenesis through at least two distinct mechanisms. Firstly, the SF preserves the infectivity of the virion through an acidic gastric environment and enables the virions to attach to their host cell through an interaction with HS. Secondly, as shown in another study, the SF stimulates the production of serotonin, causing the symptoms of diarrhea and vomiting. This points out the short fibers as key proteins, contributing not only to pathogenesis when serving as viral enterotoxins, but also to infection of host cells. The ability of enteric HAdVs to interact with HS adds to the knowledge about their distinct tropism. These results may also be useful for the design of novel antiviral drugs and to explore enteric HAdVs as tools for gene therapy and vaccine development.

## Figures and Tables

**Figure 1 viruses-13-00298-f001:**
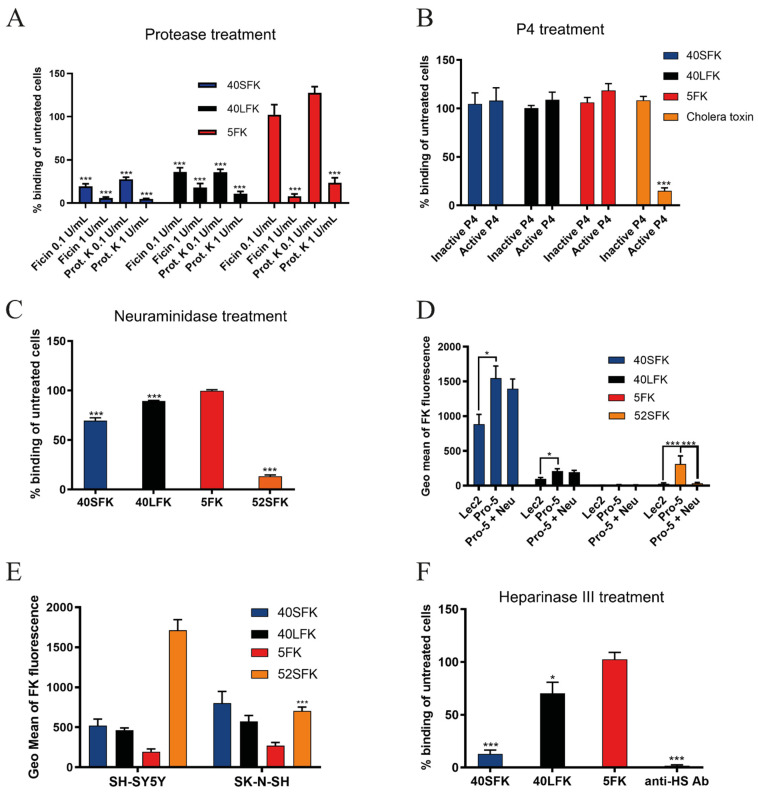
Receptor characterization for the short fiber of HAdV-F40: recombinant fiber knob binding to (**A**) A549 cells pretreated with the proteases—ficin and proteinase K—which degrade cell surface proteins; (**B**) A549 cells pretreated with P4—which inhibits glycolipid synthesis; (**C**) A549 cells pretreated with *V. cholerae* neuraminidase—which cleaves sialic acid. (**D**) CHO cells expressing or lacking sialic acid pretreated with *V. cholerae* neuraminidase (Neu). Pro-5 cells express sialic acid and Lec2 cells lack sialic acid expression. (**E**) SHSY-5Y cells expressing polysialic acid and SK-N-SH lacking polysialic acid, (**F**) A549 cells pretreated with heparinase III—which cleaves heparan sulfate. Binding was determined by geometric mean of fluorescence on a FACSLSRII flow cytometer and shown as % binding of fiber knobs on untreated cells in (**A**–**C**,**F**). All experiments were performed three times with duplicate samples in each experiment. Error bars represent mean ± SEM. * *P* < 0.05 and *** *P* < 0.001 versus control.

**Figure 2 viruses-13-00298-f002:**
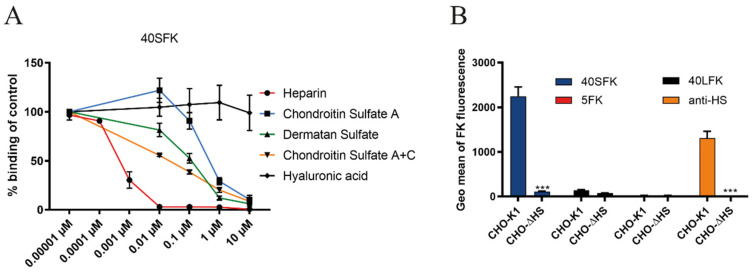
Effect of sulfation on FK binding: (**A**) 40SFK binding to A549 cells. Fiber knobs were pretreated with a broad range of concentrations of different glycosaminoglycans (GAGs). (**B**) Fiber knob binding to CHO cells expressing or lacking heparan sulfate (HS). HS expression levels are shown with an anti-HS antibody. The parental cell line CHO-K1 expresses GAGs and CHO-ΔHS specifically lacks HS but expresses other GAGs. Binding was determined by geometric mean of fluorescence on a FACSLSRII flow cytometer and shown as % binding of fiber knobs on untreated cells in (**A**). All experiments were performed three times with duplicate samples in each experiment. Error bars represent mean ± SEM. Error bars represent mean ± SEM. *** *P* < 0.001 versus control.

**Figure 3 viruses-13-00298-f003:**
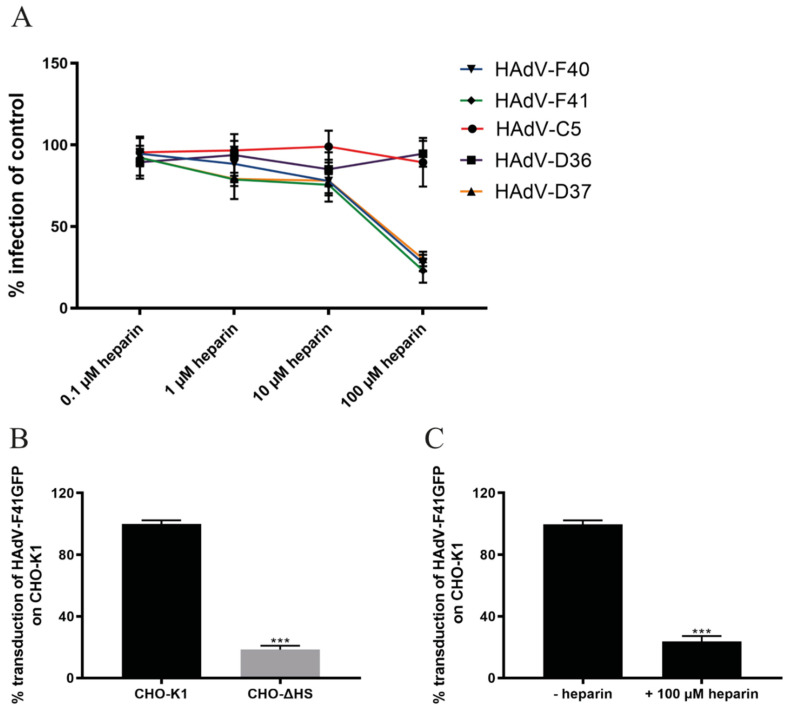
Importance of heparan sulfate on infection and transduction. (**A**) Infection of heparin pretreated HAdVs on A549 cells. The number of infected cells were quantified by immunofluorescence and shown as % infection of untreated HAdVs. (**B**) Transduction of HAdV-F41 GFP vector on CHO cells expressing or lacking heparan sulfate (HS) and (**C**) transduction of heparin pretreated HAdV-F41 GFP vector on CHO-K1 cells. The parental cell line CHO-K1 expresses glycosaminoglycans and CHO-ΔHS specifically lacks heparan sulfate but expresses other glycosaminoglycans. The number of infected/transduced cells were quantified by immunofluorescence and shown as % infection of untreated A549 cells. All experiments were performed three times with duplicate samples in each experiment. Error bars represent mean ± SEM. *** *P* < 0.001 versus control.

**Figure 4 viruses-13-00298-f004:**
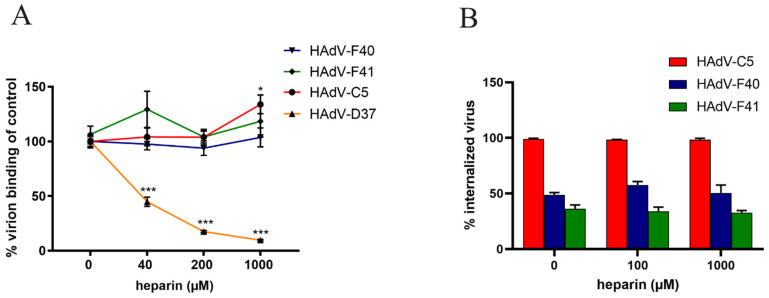
Effect of soluble heparin on virion binding and uptake in A549 cells. (**A**) ^35^S-labeled HAdV-F40 and -F41 virion binding to A549 cells. Virions were pretreated with different concentrations of soluble heparin. Binding to cells was quantified as counts per minute (CPM) by a liquid scintillation counter and represented as % binding of untreated virions. (**B**) HAdV-C5, -F40 and –F41 uptake in A549 cells. The number of intracellular virions were quantified by immunofluorescence and shown as a relative % of internalized virus in the untreated sample. The experiments were performed three times with duplicate samples. Error bars represent mean ± SEM. * *P* < 0.05 and *** *P* < 0.001 versus control.

**Figure 5 viruses-13-00298-f005:**
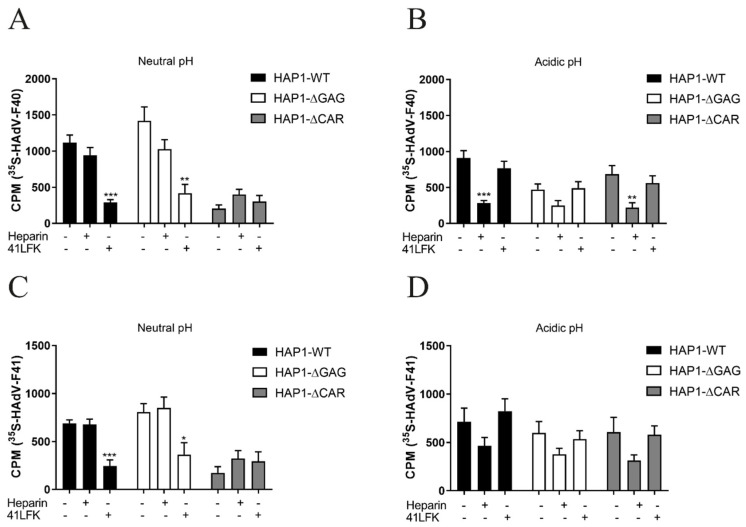
Effect of pH treatment on receptor usage: ^35^S-labeled HAdV-F40 and HAdV-F41 virion binding to HAP1 WT cells, HAP1-∆GAG cells and HAP1-∆CAR cells. The parental cell line HAP1 WT expresses glycosaminoglycans (GAGs) and CAR. HAP1-∆GAG lacks expression of GAGs, while HAP1-∆CAR lacks expression of CAR. (**A**) Untreated (neutral pH) HAdV-F40 or (**B**) acidic pH treated HAdV-F40 or (**C**) untreated (neutral pH) HAdV-F41 or (**D**) acidic pH treated HAdV-F41 were preincubated with heparin and cells were preincubated with 41LFKs before binding. Binding to cells was quantified as counts per minute (CPM) by a liquid scintillation counter. All experiments were performed three times with duplicate samples in each experiment. Error bars represent mean ± SEM. * *P* < 0.05, ** *P* < 0.01 and *** *P* < 0.001 versus control.

**Figure 6 viruses-13-00298-f006:**
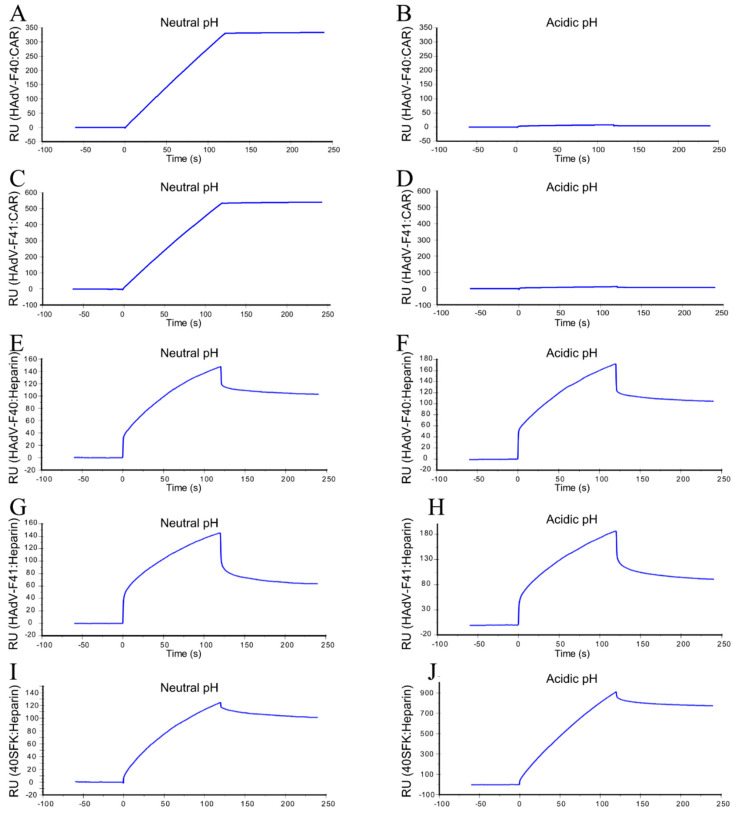
SPR analyses of HAdV-F40 and -F41 interactions with CAR and heparin and 40SFK interaction with heparin: (**A**) Untreated (neutral pH) HAdV-F40 or (**B**) acidic pH-treated HAdV-F40 or (**C**) untreated (neutral pH) HAdV-F41 or (**D**) acidic pH treated HAdV-F41 binding to immobilized CAR. (**E**) Untreated (neutral pH) HAdV-F40 or (**F**) acidic pH-treated HAdV-F40 or (**G**) untreated (neutral pH) HAdV-F41 or (**H**) acidic pH treated HAdV-F41 or (**I**) untreated (neutral pH) 40SFK or (**J**) acidic pH treated 40SFK binding to immobilized heparin. Interactions are displayed in response units (RU). All experiments were performed three times. Displayed is one representative set from each experiment.

**Figure 7 viruses-13-00298-f007:**
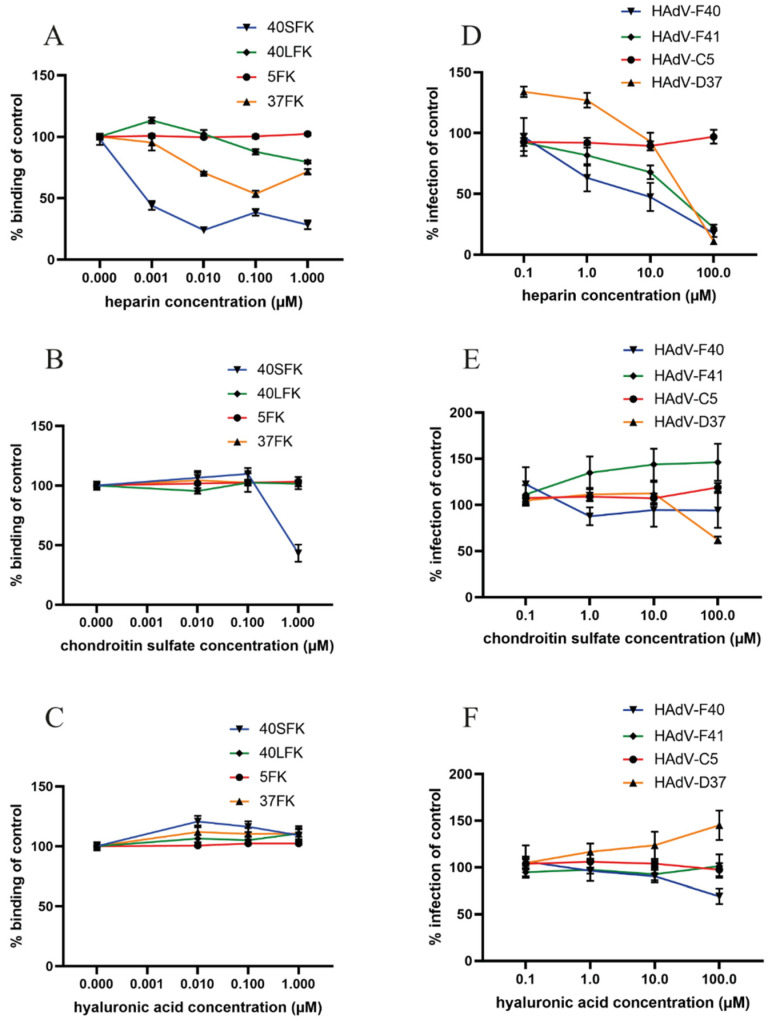
Effect of GAGs on fiber knob binding to and infection of HuTu80 cells: Binding of (**A**) heparin or (**B**) chondroitin sulfate mix or (**C**) hyaluronic acid pretreated HAdVs on HuTu80 cells. Binding was determined by geometric mean of fluorescence on a FACSLSRII flow cytometer and shown as % binding of untreated fiber knobs. Infection of (**D**) heparin or (**E**) chondroitin sulfate mix or (**F**) hyaluronic acid pretreated virions to HuTu80 cells. The number of infected cells were quantified by immunofluorescence and shown as % infection of untreated HAdVs. All experiments were performed three times with duplicate samples in each experiment.

## Data Availability

A549 cell line was provided by Alistair Kidd. HAP1-WT and HAP1-ΔCAR cell lines were purchased from Horizon Discovery (Waterbeach, UK). HAP1-ΔGAG was provided by Frank Kuppeveld. HuTu80 cell line was purchased from ATCC (Manassas, VA, USA). Pro-5, Lec2, SK-N-SH and SH-SY5Y cells were purchased from LGC Promochem (Teddington, UK).

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
