# Peer review of "Heparan Sulfate Is a Cellular Receptor for Enteric Human Adenoviruses"

_viruses, 2021, doi:10.3390/v13020298_

Round 1

Reviewer 1 Report

The authors have addressed an important issue how HAdV-40 and 41 bind to the cells and have shown that HS is a binding partner for 40 and 41. Moreover, using a clinically relevant cell line (Fig.7) they show that HS reduces binding of the HAdV-40 small fiber to the HuTu80 cells whereas the long fiber binding was not affected by HS. Elegant! The paper is well written and most of the credits should be given to the M&M section. It is very seldom nowdays to see such a detailed M&M section, big respect and thank you for that.

However, I have a few concerns with the manuscript and I hope that the authors could address them. 

Lines 245-248: the authors state that the 40LFK is substantially more sensititve to protease treatment (Fig. 1a). How can 5FK binding in the presence of low Prot.K be more than 100%? (Fig.1A). Is it technical artefact or does low Prot K treatment indeed increase 5FK binding to CAR?

Fig.1 C and 1F would benefit from a small label on the chart indicating that this data is about Neu-and Hepa-treatments. Then it will be easier to catch the what kind of treatments were done without the need to read the figure caption in details. Something what you have on Figs1A, 1B, 1D.

In the text (lanes 293-312) the authors write that different GAGs (CS, HS and DS) have different effects on 40SFK binding and point towards the Fig.2A. However, the Fig.2A X-axis and the caption text do not fit since the X-axis tells about HS concentration effect but the caption text is about using different GAGs. This has to be made more understandable to the reader. Also the authors should comment why the HAdV-D36 is not behaving as the D37, F40, F41. NB! See also my comment below!

Fig.2 and Fig.3 have exactly the same data! The figures have been duplicated, hence the reviewer does not even bother to figure out what is right and what is wrong since it does not make any sense. These things happen, but a better proofreading should have been done by the corresponding author. Funny enough, it is stated that all the authors have read the manuscript and no one noticed the duplication.

Author Response

Point 1: Lines 245-248: the authors state that the 40LFK is substantially more sensititve to protease treatment (Fig. 1a). How can 5FK binding in the presence of low Prot.K be more than 100%? (Fig.1A). Is it technical artefact or does low Prot K treatment indeed increase 5FK binding to CAR?

Response 1: We appreciate the reviewer’s concern regarding the 25-30% increase in 5FK binding to A549 cells after low proteinase K treatment. We speculate that this increase in binding with low proteinase K treatment could be because of incomplete or partial removal of protein factors from the cell surface revealing novel binding sites for 5FK. We did not see this effect after treatment with 1% proteinase K since at this concentration, the removal of cell surface proteins was complete. If the reviewer/editor insists, a sentence addressing this can be added to the manuscript as follows:  

Both 5FK and 40LFK bind CAR, but we observed a remarkable difference in sensitivity to protease treatment between the two, with 40LFK being substantially more sensitive than 5FK. In addition, we observed an increase in 5FK binding at low proteinase K concentrations that could potentially result from partial removal of protein factors from the cell surface revealing novel binding sites for 5FK. We did not evaluate this result further as it did not fall within the scope of this study.

Point 2: Fig.1 C and 1F would benefit from a small label on the chart indicating that this data is about Neu-and Hepa-treatments. Then it will be easier to catch the what kind of treatments were done without the need to read the figure caption in details. Something what you have on Figs1A, 1B, 1D.

Response 2: We fully agree with the reviewer and have added a label to figures 1C and 1F (please see figure in attached file).

Point 3: In the text (lanes 293-312) the authors write that different GAGs (CS, HS and DS) have different effects on 40SFK binding and point towards the Fig.2A. However, the Fig.2A X-axis and the caption text do not fit since the X-axis tells about HS concentration effect but the caption text is about using different GAGs. This has to be made more understandable to the reader. Also the authors should comment why the HAdV-D36 is not behaving as the D37, F40, F41. NB! See also my comment below!

Response 3: We profusely apologise about inserting the wrong figure in place of Figure 2, making it difficult for the reviewer to follow our results. We hope that with the right figure in place, this part of the result section will be more understandable.

Although HAdV-D36 and HAdV-D37 are closely related and belong to the same species of HAdVs, HAdV-D36, HAdV-D37 and enteric HAdV-F40 and -F41 infect different cell types and cause different diseases, which could indicate a difference in receptor usage. Since HAdV-D37 has been shown to bind HS as a decoy receptor, addition of soluble heparin reduces the infection. However, in our studies heparin does not affect HAdV-D36 infection in A549 cells, indicating that HS is not a receptor or a decoy receptor for this virus. Studies on receptor usage by HAdV-D36 are currently ongoing.

Point 4: Fig.2 and Fig.3 have exactly the same data! The figures have been duplicated, hence the reviewer does not even bother to figure out what is right and what is wrong since it does not make any sense. These things happen, but a better proofreading should have been done by the corresponding author. Funny enough, it is stated that all the authors have read the manuscript and no one noticed the duplication.

Response 4: Again, we are extremely apologetic and regretful about this mistake and have rectified it (please see figure in attached file).

Reviewer 2 Report

Dear Sirs,
I enjoyed reading your manuscript and recommend acceptance as is/minor revision. Despite of that I have a few comments:
1. Honestly, prior to our investigation there were reports suggesting that HaD5 though KKTK domain binds to Heparan sulfate receptor, if Im not mistaken. You investigation suggests "Nevertheless, we observed a specific reduction of infection by heparin that could not be seen with CS or HA, neither could it be explained by a general negative effect on the cells since no effect was observed on the control viruses HAdV-C5 and HAdV-D36.' However, Fig 4 of " Non-heparan sulfate GAG-dependent infection of cells using an adenoviral vector with chimeric fiber conserving its KKTK motif"
2. Investigation by Ulasov et all 2007 and earlier by WU et all 2002 and then Rein et all 2004 suggested that 7 polylisine domains (AdWT-pK7 fiber) also binds to HS receptor, and knob charge provides foundation to binding to HS. Have you measured any charge for HAd41 short fiber? Will be in this case possibility of binding for two different fibers( AdWT-PK7) and Had41 short fiber to different types of HS domains/receptors?

Author Response

Point 1: Honestly, prior to our investigation there were reports suggesting that HaD5 though KKTK domain binds to Heparan sulfate receptor, if Im not mistaken. You investigation suggests "Nevertheless, we observed a specific reduction of infection by heparin that could not be seen with CS or HA, neither could it be explained by a general negative effect on the cells since no effect was observed on the control viruses HAdV-C5 and HAdV-D36.' However, Fig 4 of " Non-heparan sulfate GAG-dependent infection of cells using an adenoviral vector with chimeric fiber conserving its KKTK motif"

Response 1: We value the reviewer’s concern regarding the KKTK motif on the adenoviral vector used. In figure 3, we have used a HAdV-F41 GFP vector, which is not a chimeric vector but wholly based on HAdV type 41 including all the capsid proteins – hexon, penton base, fiber shaft and fiber knob. Since this vector does not contain any capsid proteins from HAdV-C5, there is no KKTK motif present that could affect the results in this experiment. In figure 4, only wild type HAdVs were used to measure binding and uptake on A549 cells in the presence of heparin.

Point 2: Investigation by Ulasov et all 2007 and earlier by WU et all 2002 and then Rein et all 2004 suggested that 7 polylisine domains (AdWT-pK7 fiber) also binds to HS receptor, and knob charge provides foundation to binding to HS. Have you measured any charge for HAd41 short fiber? Will be in this case possibility of binding for two different fibers( AdWT-PK7) and Had41 short fiber to different types of HS domains/receptors?

Response 2: The predicted pI values of HAdV-F40 and -F41 SFs are 7.78 and 9.13, respectively. The SFs have a higher pI value than most HAdV fibers, which have an average pI around 5.0 and 6.0, except for species D (HAdV-D8 pI= 9.42 and HAdV-D37 pI= 9.56) (1). In agreement with the studies mentioned above, the knob charge could provide the foundation for binding to HS since both HAdV-D8 and –D37 also bind HS. We have seen that treatment of cells with sodium chlorate to prevent de novo sulfation of carrier molecules, reduces infection by both HAdV-F40 and -F41, indicating the need of negatively charged sulfate for efficient binding. The 7 polylysine domain inserted in the AdWT-PK7 knob will provide a local positive charge that will likely bind to highly sulfated domains on HS. There is a chance that the 41SF and 40SF also binds to these domains, but more studies will be needed to address this question in more detail.

Round 2

Reviewer 1 Report

The authors  have addressed my concerns and have uploaded the correct Figure 2. I would like that the authors include their suggested text ("Both 5FK and 40LFK bind CAR, but we observed a remarkable difference in sensitivity to protease treatment between the two, with 40LFK being substantially more sensitive than 5FK. In addition, we observed an increase in 5FK binding at low proteinase K concentrations that could potentially result from partial removal of protein factors from the cell surface revealing novel binding sites for 5FK. We did not evaluate this result further as it did not fall within the scope of this study.") into the manuscript. Otherwise it is an interesting paper, well done!

Author Response

Point 1: The authors  have addressed my concerns and have uploaded the correct Figure 2. I would like that the authors include their suggested text ("Both 5FK and 40LFK bind CAR, but we observed a remarkable difference in sensitivity to protease treatment between the two, with 40LFK being substantially more sensitive than 5FK. In addition, we observed an increase in 5FK binding at low proteinase K concentrations that could potentially result from partial removal of protein factors from the cell surface revealing novel binding sites for 5FK. We did not evaluate this result further as it did not fall within the scope of this study.") into the manuscript. Otherwise it is an interesting paper, well done!

Response 1: We thank the reviewer for their appreciation of our work and for accepting our responses. We have included the following sentence at line 244:

In addition, we observed an increase in 5FK binding at low proteinase K concentrations that could potentially result from partial removal of protein factors from the cell surface revealing novel binding sites for 5FK.